# Detection and Positioning of *Camellia oleifera* Fruit Based on LBP Image Texture Matching and Binocular Stereo Vision

Xiangming Lei [1], Mingliang Wu [1], Yajun Li [1,2], Anwen Liu [1], Zhenhui Tang [1], Shang Chen [1] and Yang Xiang [1,*]

1   College of Mechanical and Electrical Engineering, Hunan Agriculture University, Changsha 410128, China
2   Intelligent Equipment Research Center, Beijing Academy of Agriculture and Forestry Sciences, Beijing 100097, China
*   Correspondence: xy@hunau.edu.cn

**Abstract:** To achieve the rapid recognition and accurate picking of *Camellia oleifera* fruits, a binocular vision system composed of two industrial cameras was used to collect images of *Camellia oleifera* fruits in natural environments. The YOLOv7 convolutional neural network model was used for iterative training, and the optimal weight model was selected to recognize the images and obtain the anchor frame region of the *Camellia oleifera* fruits. The local binary pattern (LBP) maps of the anchor frame region were extracted and matched by using the normalized correlation coefficient template matching algorithm to obtain the positions of the center point in the left and right images. The recognition experimental results showed that the accuracy rate, recall rate, *mAP* and $F_1$ of the model were 97.3%, 97.6%, 97.7% and 97.4%. The recognition rate of the *Camellia oleifera* fruit with slight shading was 93.13%, and the recognition rate with severe shading was 75.21%. The recognition rate of the *Camellia oleifera* fruit was 90.64% under sunlight condition, and the recognition rate was 91.34% under shading condition. The orchard experiment results showed that, in the depth range of 400–600 mm, the maximum error value of the binocular stereo vision system in the depth direction was 4.279 mm, and the standard deviation was 1.142 mm. The detection and three-dimensional positioning accuracy of the binocular stereo vision system for *Camellia oleifera* fruits could basically meet the working requirements of the *Camellia oleifera* fruit-picking robot.

**Keywords:** binocular stereo vision; LBP map; 3D positioning; image matching; *Camellia oleifera* fruit





## 1. Introduction

*Camellia oleifera* (*C. oleifera*) fruit is a type of *Camellia* oil raw material that has high nutritional value [1]. By the end of 2020, the *C. oleifera* fruit planting area in China reached 4.451 million hectares. C. oleifera fruit is mostly planted in the hilly areas of southern China, where mechanized harvesting is difficult. Furthermore, *C. oleifera* fruit has a flower and fruit synchronization characteristic, so it is easy to damage the buds and affect the yield of the following year when traditional harvesting machinery is used for harvesting. The precise picking of *C. oleifera* fruits using picking devices such as manipulators can improve harvesting efficiency and avoid bud damage as much as possible, and the accurate identification and precise positioning of *C. oleifera* fruits in the field are the basis for achieving precise picking.

With the development of target detection algorithms and improvements in computer performance, the mainstream method used by scholars for *C. oleifera* fruit detection research in natural environments is a deep convolution neural network, which has good effect [2–5], and among which the YOLOv7 network model has good robustness in detecting *C. oleifera* fruits in natural environments [6,7].

Binocular stereo vision and RGB-D (red, green, blue and depth) vision are the most commonly used three-dimensional spatial information acquisition technologies at present [8]. Li et al. integrated an efficient cost calculating measure—AD-Census—with the adaptive

support weight (ASW) approach to improve the accuracy of binocular stereo vision for plant 3D imaging [9]. Gao et al. used the HSV (hue, saturation, value) color space to segment a pitaya image and obtained the 3D coordinates of pitaya through SURF (speeded-up robust features) matching binocular vision, with an average positioning error of 6 mm [10], but the SURF algorithm required a large amount of computation to extract feature points. Hsieh et al. used the R-CNN (region with convolutional neural network features) network model to recognize ripe tomato in a greenhouse, and the depth calculation algorithm of ZED mini binocular camera to obtain depth information, and the results showed that the average depth localization error was 6.7 mm [11]. Liu et al. identified the position of pineapple in an image based on the improved YOLOv3 model and realized the localization of the target pineapple by matching the pineapple anchor block diagram with template matching, with an average absolute error of 24.414 mm at a distance of 1.7–2.7 m [12]. Tang et al. used a ZED2 binocular camera to realize the detection of *C. oleifera* fruits based on the improved YOLOv4-tiny model, and used the normalized cross-correlation method based on gray level to search the best match of the center point for positioning in one-dimensional straight line, which reduced the range of the match and improved matching speed, and the positioning accuracy of the orchard experiment was 23.568 mm [13].

The RGB-D camera is widely used because of its mature technology, perfect program interface, and easy to obtain depth information [14–18]. Lin et al. used the RGB-D sensor to detect guava fruit and estimate its pose in real outdoor conditions [19]. Li et al. used the YOLOv3 network model to detect tea buds, fused the RGB-D camera depth image and RGB image, and achieved an average depth error at a distance of 1 m of less than 5 mm [20]. Yu et al. proposed a mature pomegranate fruit detection and positioning method based on an improved F-PointNet and 3D clustering method. Combined with the 3D point cloud data of an RGB-D camera, the positioning error was within 5 mm [21]. Lin constructed a mask R-CNN network with an Inception_v2 network as feature extraction network and used an Intel D435i structured light camera to locate the three-dimensional position of *C. oleifera* fruits. In the experiment, the positioning error was 6.1 mm [22]. Xu et al. used a camera and laser radar for joint calibration, fusing the image and point cloud data. The experimental positioning error was within 10 mm [23]. The ranging method was the same as that of the RGB-D camera, which was based on the active measurement of 3D space by structured light to obtain the depth information.

Although RGB-D camera has good positioning accuracy, strong illumination easily affects the performance of the camera sensor in a natural environment [24], and binocular camera has strong anti-interference ability to light, it is applicable in an orchard's natural environment. To ensure the imaging quality and positioning accuracy, a *C. oleifera* fruit detection and positioning method based on a binocular stereo vision system is proposed in this study to solve the above problems. The main contributions of our work are summarized as follows:

- Two industrial cameras are set up to form a binocular stereo vision system for recognizing and locating *C. oleifera* fruits.
- A robust detection model is obtained by using the YOLOv7 neural network to train the dataset of *C. oleifera* fruits.
- A binocular vision localization algorithm based on image matching with local binary patterns (LBP) maps is proposed.

## 2. Materials and Methods

### 2.1. Experimental Materials and Acquisition Equipment

The *C. oleifera* fruit images used in this study were collected from a *C. oleifera* fruit planting base in Jinjing Town, Changsha County Changsha City, Hunan Province. The shooting time included morning, noon, and evening. A total of 1600 images were taken from different angles, such as forward light and reverse light, respectively, as the original dataset.

The built binocular stereo vision system is shown in Figure 1. Two Hikvision Gigabit Ethernet planar array industrial cameras were used as image acquisition cameras, model

MV-CA060-10GC (Hangzhou Haikang Robot Co., Ltd., Hangzhou, China), with a resolution of 3072 × 2048 and a lens focal length of 6 mm. A Hikvision gigabit POE (power over ethernet) (Hangzhou Haikang Robot Co., Ltd., Hangzhou, China) switch was adopted for use as network transmission equipment. The image processing terminal adopted a Xiaomi 2019 laptop(Xiaomi Corporation, Peking, China), which was configured with Intel Core i7 9750H, a NVIDIA GeForce RTX 2060 graphics card(NVIDIA Corporation, Santa Clara, CA, USA), and 16 GB of memory(Kingston, Fountain Valley, California, USA); Anaconda 3.6 was selected to construct the training and detection environment; Pytorch 1.10, CUDA 11.3 and Opencv-Python 4.5 were used to realize the proposed detection and localization algorithm in Python language (Python 3.10).

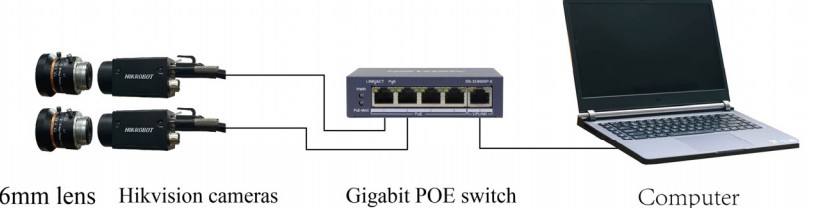

**Figure 1.** Binocular stereo vision system.

## 2.2. Object Detection Algorithms of C. oleifera Fruit Based on YOLOv7

The YOLO series of object detection algorithms are based on the regression idea of deep learning and use a one-stage model to simultaneously predict the positions and categories of multiple bounding boxes to quickly detect objects.

The overall network structure of YOLOv7 can be divided into three parts: input (input layer), backbone (feature extraction network), and neck and head (detection head layer). In the structure of YOLOv7, the ELAN (efficient layer aggregation network) structure is mainly added to enhance the network to learn the characteristics of *C. oleifera* fruits in different characteristic images, and to improve the parameter use and calculation. The neck and head layer adopts the current popular RepConv (reparametric convolution) structure and fuses different convolution layers into a convolution module by fusing the reparameterization residual structure to improve the training accuracy and make the inference speed faster [25].

In this study, a YOLOv7 target detection framework was selected to iterative training *C. oleifera* fruits data sets, and the optimal weight model for detection in real time was obtained through 300 rounds of training [26]. In the preprocessing stage of real-time detection, the resolution of *C. oleifera* fruit image was converted from 3072 × 2048 to 768 × 512 to improve the detection speed.

## 2.3. Extraction of Center Point Coordinates of C. oleifera Fruit

According to the fact that the LBP operator has gray invariance and insensitivity to illumination change [27], the LBP maps are used for image matching, which can effectively improve the robustness of the positioning algorithm in natural environments; considering the real-time requirement of the algorithm in practical application, the basic LBP operator with the shortest processing time was selected as the operator type of LBP map extraction. In this study, the center point of the anchor frame region of the left camera image (the left image) is taken as the center point of the *C. oleifera* fruit, and the center point of the *C. oleifera* fruit in the right camera image (the right image) is obtained using image template-matching of the LBP maps in the center region of the anchor frame, in order to display more image details and increase the accuracy of template matching, the process was carried out at the original resolution of 3072 × 2048, with the processing flow shown in Figure 2. The rectangular anchor frame regions of *C. oleifera* fruits can be obtained using the detection model to detect the fruits in the image, which are approximately the circumscribed rectangular frames of the contour of *C. oleifera* fruits. The anchor frame regions corresponding to the same *C. oleifera* fruit in the left and right images need to be matched through the multiple constraint conditions combining sequence consistency and

epipolar constraint [28]. Considering that the anchor frames of the same *C. oleifera* fruit in the left and right images through the detection model may have deviation in the vertical direction, the matching range of the epipolar constraint principle is expanded to 5 pixel points above and below, as shown in Figure 2a. To reduce the influence of the complex background on the subsequent image processing, image cropping is performed on each anchor frame region, as shown in Figure 2b. Grayscale processing is performed on the anchor frame image, as shown in Figure 2c. The central point of the anchor frame image is taken as the center, the LBP map of a rectangular region with the length and the width of one quarter of the left anchor frame image is extracted as the template image to ensure that the template image has stronger correlation with an image to be matched, and to improve the accuracy of template matching. The LBP map of the rectangular region with the length and the width of half of the right anchor frame image is extracted as the region to be matched, as shown in Figure 2e. Normalized correlation coefficient template matching is performed on the LBP maps to obtain a region with the maximum correlation coefficient in the right anchor frame, and the center point of the region in the right anchor frame image is the corresponding matching point of the center point of the left anchor frame image, as shown in Figure 2f.

### 2.4. Binocular Stereo Vision Ranging Principle

Binocular stereo vision simulates human binocular vision and calculates the distance between the object through the parallax produced by the left and right cameras when the cameras shoot the object [29]. The coordinates of *C. oleifera* fruits with different distances in the pixel plane of the left and right cameras are different. The closer the distance is, the greater the difference of the projection of the *C. oleifera* fruits in the image plane of the left and right cameras is, and conversely, the difference is smaller, as shown in Figure 3.

In Figure 3: $P$ is the center point of the *C. oleifera* fruit; $b$ is the fixed distance between the left and right cameras, i.e., the baseline; to calculate conveniently, the pixel coordinate systems $C_L$ and $C_R$ of the left and right cameras are transformed to be between the optical center of the cameras and the *C. oleifera* fruit by translation and rotation transformation, the distance between the plane of the transformed coordinate system and the optical center is kept as the focal length $f$, and the plane of the transformed coordinate system is parallel to the original imaging plane; $p_1$ and $p_2$ are the projection points of the same *C. oleifera* fruit in the imaging planes of the left and right cameras; $(x_l, y_l)$ and $(x_r, y_r)$ are the coordinates of $p_1$ and $p_2$ in the pixel coordinate systems $C_L$ and $C_R$, respectively, and are also the coordinates of the center point of the *C. oleifera* fruit in the left and right images obtained after the image template matching of the LBP maps; the vertical distance between the central point $P$ of the *C. oleifera* fruit and the plane of the coordinate system formed by the optical centers $O_L$ and $O_R$ of the left and right cameras is $Z$; and $(X, Y, Z)$ is the three-dimensional space coordinate of $P$ in the camera coordinate system established by taking the optical center of the left camera as an origin.

According to the principle of triangulation, the calculation equation is as follows:

$$\frac{Z}{f} = \frac{X}{x_l} = \frac{X - b}{x_r} \tag{1}$$

The three-dimensional coordinates of the center point of the *C. oleifera* fruit are:

$$Z = \frac{fb}{x_l - x_r} \tag{2}$$

$$X = \frac{x_l b}{x_l - x_r} \tag{3}$$

$$Y = \frac{y_l b}{x_l - x_r} \tag{4}$$

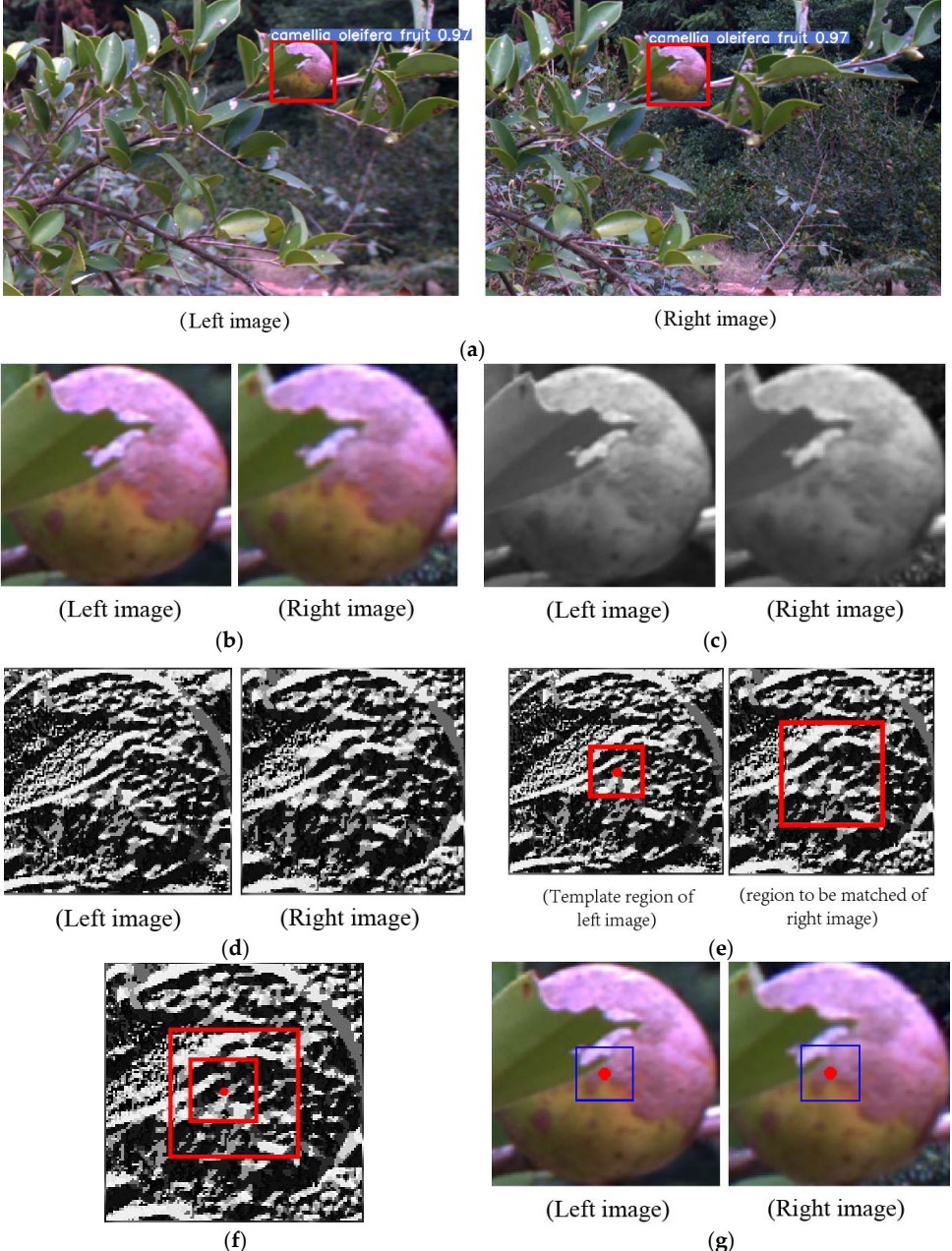

**Figure 2.** Center point coordinate extraction process: (**a**) match the anchor frame regions(The red box is the location of *C. oleifera* fruit in the recognition map of *C. oleifera* fruit detection model); (**b**) cut the anchor frame regions; (**c**) grayscale image processing; (**d**) local binary patterns (LBP) maps; (**e**) set the template region and the region to be matched (The red box is the LBP matching region extracted in the Left image and Right image, and the red dot is the center point of the *C. oleifera* fruit determined in the Left image); (**f**) result of template matching (The red box is the area of LBP image texture matching, and the red dot is the center point of *C. oleifera* fruit obtained after matching); (**g**) extract the center point coordinates of binocular images (The blue box is the area of LBP image texture matching, and the red dot is the center point of *C. oleifera* fruit obtained after matching).

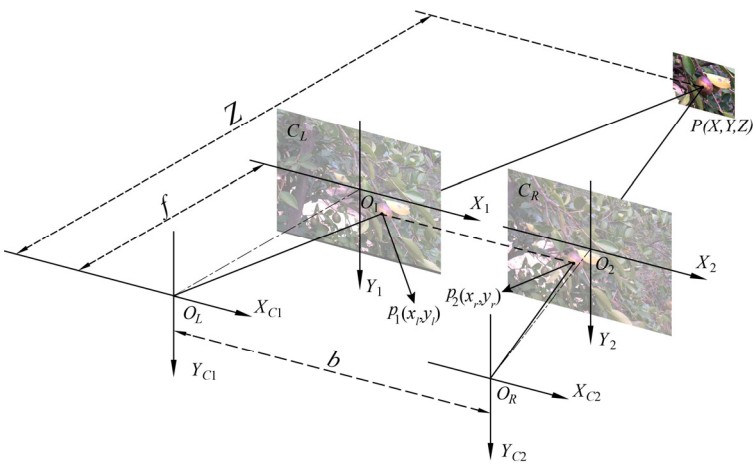

**Figure 3.** Binocular stereo vision ranging schematic diagram.

### 3. Results

*3.1. C. oleifera Fruit Recognition Experiment*

In this study, a dataset of 1600 images was divided into a training set, a verification set, and a test set according to the proportion of 7:1.5:1.5, and the optimal weight model after iterative training was selected as the YOLOv7 algorithm real-time detection model for *C. oleifera* fruit recognition. In addition, the confidence value of target detection was set to 0.7 to reduce the false detection rate of the algorithm, and the precision *P*, recall *R*, average precision *mAP*, and comprehensive evaluation index $F_1$ of the detection model were selected as the indicators for evaluating the performance of the model [30]. The calculation equation of $F_1$ is as shown in Equation (5):

$$F_1 = \frac{2PR}{P+R} \tag{5}$$

The performance evaluation index data of the detection model obtained from the verification of the test set are shown in Table 1.

**Table 1.** Model performance evaluation metrics data.

| *P* (%) | *R* (%) | *mAP* (%) | $F_1$ (%) | *t* (s/pic) |
|---|---|---|---|---|
| 97.3 | 97.6 | 97.7 | 97.4 | 0.021 |

Note: *t* is the average detection speed of the model.

To further verify the recognition effect of the detection model on *C. oleifera* fruits in complex natural environments, the experiment carried out recognition testing for different illumination and occlusion conditions, which are the two main factors affecting the recognition effect [31]. In the experiment, 40 images of *C. oleifera* fruits with different degrees of occlusion were selected and divided into no occlusion, slight occlusion and severe occlusion (the fruits occluded by other fruits or leaves exceed 50%), according to the degree of occlusion. In addition, 20 images of *C. oleifera* fruits with large differences in illumination intensity caused by shooting at different angles were divided into strong light and back light, according to the illumination intensity. Furthermore, the above 60 images were selected from the training set and the verification set.

The statistical results of the experiment are shown in Tables 2 and 3.

According to the results in Table 2, the detection model can accurately recognize *C. oleifera* fruits in the case of no occlusion and slight occlusion, with the recognition rate reaching 99.29% and 93.13%. In the case of severe occlusion, the recognition rate decreased to 75.21%, and most of the unrecognized fruits were occluded by more than 70%. The detection model can accurately recognize fruits within the uncovered range, as shown in Figure 4b.

**Table 2.** Recognition result statistics of *C. oleifera* fruits under occlusion conditions.

| The Degree of Occlusion | Number of Fruits Detected | Number of Actual Fruits | Recognition Rate |
|---|---|---|---|
| No occlusion | 141 | 142 | 99.29% |
| Slight occlusion | 95 | 102 | 93.13% |
| Severe occlusion | 88 | 117 | 75.21% |

**Table 3.** Recognition results statistics of *C. oleifera* fruits under different illumination conditions.

| Light Conditions | Number of Fruits Detected | Number of Actual Fruits | Recognition Rate |
|---|---|---|---|
| Sunlight | 126 | 139 | 90.64% |
| Shading | 95 | 104 | 91.34% |

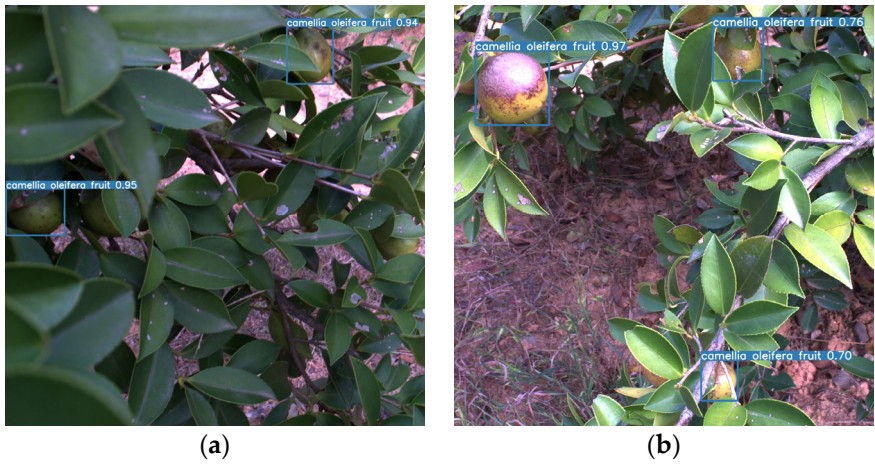

(a)  (b)

**Figure 4.** Recognition effect of *C. oleifera* fruits with different occlusion: (**a**) slight occlusion; (**b**) severe occlusion.

Table 3 shows the detection results of the model under the different illumination conditions of sunlight and shading, reaching 90.64% and 91.34%, respectively, with the recognition effect of some images shown in Figure 5a,b.

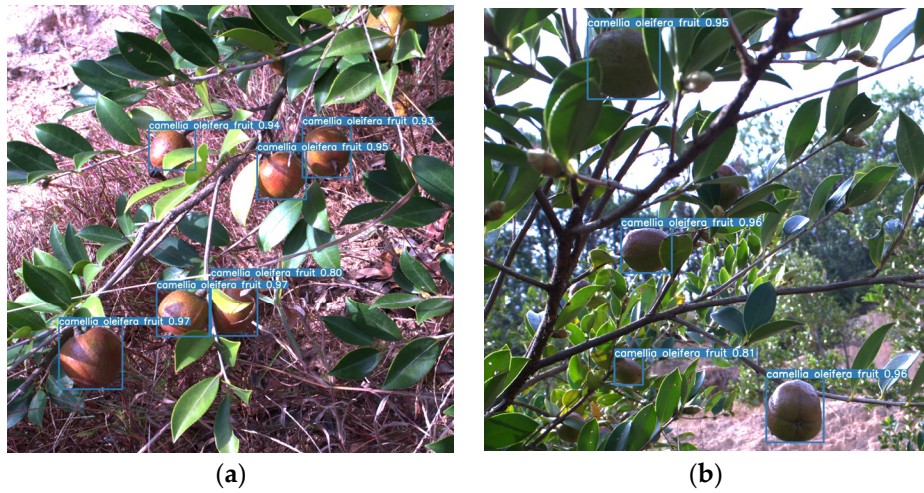

(a)  (b)

**Figure 5.** Recognition effect of *C. oleifera* fruits with different occlusion: (**a**) front light; (**b**) back light.

### 3.2. Experiment on Indoor Positioning Accuracy of Binocular Stereo Vision

To accurately evaluate the 3D positioning accuracy of *C. oleifera* fruits with the binocular stereo vision system, an indoor experiment platform was built using an industrial robot.

The indoor experiment platform mainly consisted of an industrial robot with a six-axis mechanical arm as the core, a teach pendant, the binocular stereo vision system and a mobile bracket, as shown in Figure 6. The model of the industrial robot was Borunte 0707A, and it had been calibrated before delivery. The motion accuracy was ±0.03 mm, the arm span was 700 mm, and the probe was fixed at the end of the industrial robot. The *C. oleifera* fruit was fixed on the top of the mobile bracket. The probe was accurately moved to touch the mark point of the *C. oleifera* fruit vertically by operating the teach pendant. The three-dimensional space coordinate of the current tail end of the probe under the coordinate system of the mechanical arm was displayed on the teach pendant at that moment. The coordinates (data compensation had been performed in combination with the radius of the *C. oleifera* fruit) were recorded as the real values. The three-dimensional coordinates of the center point of the *C. oleifera* fruit were measured using the binocular vision system with the *C. oleifera* fruit detection and positioning algorithm and were taken as the measured values and converted to the robot coordinate system through coordinate transformation. A plurality of groups of experimental data were obtained by changing the three-dimensional position of the *C. oleifera* fruit by moving the mobile bracket. The errors between the real values and the measured values in the $X$-, $Y$- and $Z$-axis directions were $\Delta X$, $\Delta Y$ and $\Delta Z$, respectively, and $\Delta E$ was used as the 3D positioning comprehensive error of binocular stereo vision and the standard for measuring the positioning accuracy [31]:

$$\Delta E = \sqrt{\Delta X^2 + \Delta Y^2 + \Delta Z^2} \tag{6}$$

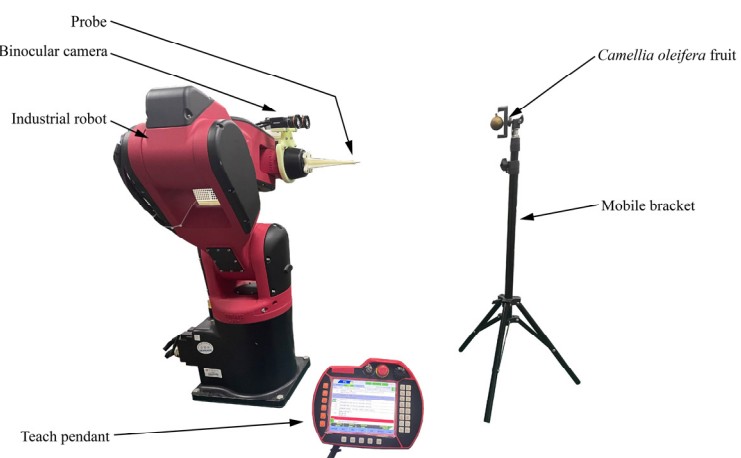

**Figure 6.** Indoor experiment platform.

The pixel coordinate system of a binocular camera can be expressed through coordinate transformation because of the linear relationship between it and the 3D space coordinate system of the mechanical arm [32]. In this study, the origin of the binocular camera coordinate system was set at the optical center of the left camera, and the origin of the robot coordinate system was the projection point of the first axis rotation center on the bottom surface of the base. The coordinates conversion equation was expressed using a homogeneous matrix, as shown in Equation (7):

$$\begin{bmatrix} X_C \\ Y_C \\ Z_C \\ 1 \end{bmatrix} = \begin{bmatrix} R & t \\ 0^T & 1 \end{bmatrix} \begin{bmatrix} X_W \\ Y_W \\ Z_W \\ 1 \end{bmatrix} \tag{7}$$

where $(X_C, Y_C, Z_C)$ refers to the three-dimensional coordinates of the marked points of the *C. oleifera* fruits in the camera coordinate system; $(X_W, Y_W, Z_W)$ denotes the three-dimensional coordinates of the marked points of the *C. oleifera* fruit in the robot coordinate system; and *R* and *t* represent the rotation matrix and the translation matrix of the transformed coordinate system, respectively.

The three-dimensional space coordinates of the center point of the *C. oleifera* fruit in the robot coordinate system were obtained by calculating the coordinates in the camera coordinate system according to Equations (2)–(4) and inputting the coordinates into Equation (7).

In the camera coordinate system, the 3D positioning accuracy of the spatial points within the field of view of the binocular camera was tested in the depth planes of 400 mm, 450 mm, 500 mm, and 550 mm, respectively. In the experiment, the test points were uniformly set along the *X*-axis and *Y*-axis in each depth plane, and the distance between adjacent points was 50 mm. Within the depth plane of 400 mm, the field of view in the *X*-axis direction was −140 mm to 90 mm, and the field of view in the *Y*-axis direction was −120 mm to 50 mm, with a total of 24 test points set accordingly. With the increase in the depth distance, the field of view was correspondingly expanded, and the number of spatial points set was correspondingly increased. When the depth distance *Z* was 550 mm, the field of view in the *X*-axis direction was −210 mm to 140 mm, and the field of view in the *Y*-axis direction was −180 mm to 120 mm. A total of 72 test points were set. The error statistic chart was plotted by inputting the data of these test points into Equation (7) for calculation, as shown in Figure 7. The average elapsed time of the identification and positioning algorithms in the test process as shown in Table 4.

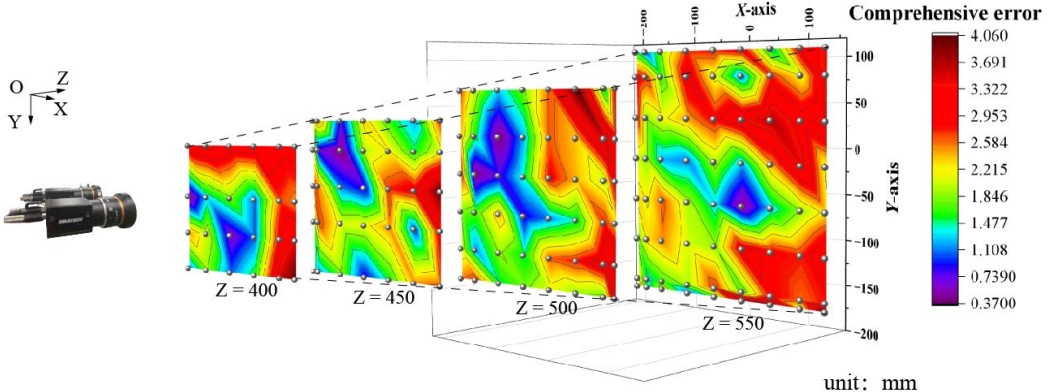

**Figure 7.** Statistical map of indoor three-dimensional positioning error.

**Table 4.** Speed statistics at different stages of recognition and positioning algorithms.

|  | **Preprocessing Stage** | **Recognition Stage** | **Positioning Stage** |
|---|---|---|---|
| **Average elapsed time** | 0.012 s/pic | 0.021 s/pic | 0.028 s/fruit |

Note: The average elapsed time of matching a single *C. oleifera* fruit in the positioning stage was 0.028 s.

It can be seen from Figure 7 that the three-dimensional positioning accuracy of the binocular stereo vision system was relatively stable within the depth range of 400–550 mm. With the deepening of measurement depth, the visual field range increases correspondingly, the error increases gradually and the points with large errors increases. Within the depth range of 400–550 mm, the maximum comprehensive error was 4.06 mm, the minimum comprehensive error was 0.37 mm, and the average comprehensive error was 2.306 mm.

*3.3. Experiment in Natural Orchard Environment*

To further verify the performance of the binocular stereo vision system in a natural environment, a detection and positioning experiment was carried out in a *C. oleifera*

fruits planting base. Considering that it is difficult to accurately measure the actual three-dimensional spatial coordinates of *C. oleifera* fruits in a natural orchard environment, a portable experimental device for measuring the absolute depth value of the target object in an orchard environment was developed (Figure 8). The laser rangefinder model in the device was a KOMAX K-50(Komax Shanghai Co., Ltd., Shanghai, China), the measurement accuracy was ±1 mm, and the range was 50 m. The precise depth of *C. oleifera* fruits can be obtained using a laser rangefinder and compared with the measurement values obtained by the binocular stereo vision system. In the experiment, the red light spot shot using the device at a certain distance from the target *C. oleifera* fruit was aligned with the center of the *C. oleifera* fruit. In addition, the experimental data of the laser rangefinder and the binocular stereo vision system were recorded, respectively, after the data were stable. The schematic diagram of the distance measuring device is shown in Figure 9. For comparison and analysis convenience, the measurement data of the laser rangefinder were transformed into the camera coordinate system. In Figure 9, $L_{real}$ was measured by the laser rangefinder as the real value, and $\Delta d$ was the distance between the binocular camera coordinate system and the datum surface of the laser rangefinder in the depth direction. Equation (2) was used to calculate the depth $L_{measure}$ of the center point of *C. oleifera* fruit measured in the binocular camera coordinate system, and the error $e$ between the real value and the measurement value was calculated as follows:

$$e = \mid L_{real} - \Delta d - L_{measure} \mid \tag{8}$$

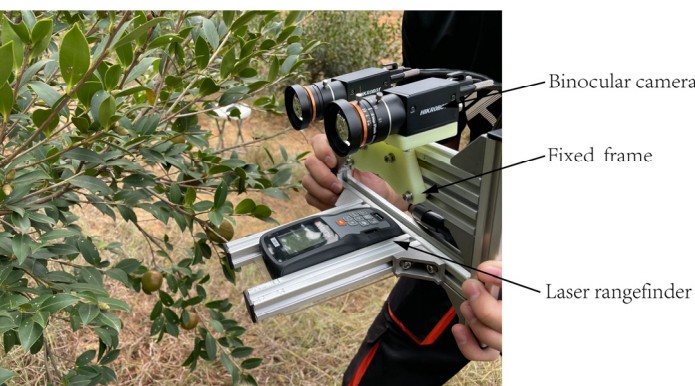

**Figure 8.** Distance measuring device in the orchard.

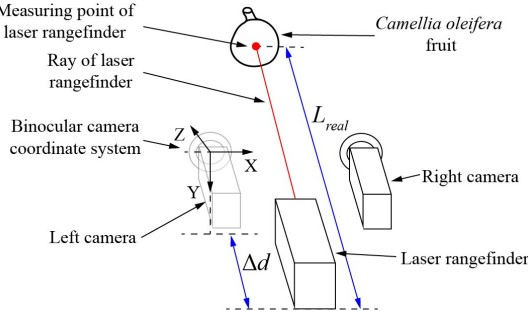

**Figure 9.** Schematic diagram of the distance measuring device.

The recognition and binocular image matching experiments were carried out on the *C. oleifera* fruits under strong light and weak light conditions, respectively, with the results shown in Figures 10 and 11. For the center point of the *C. oleifera* fruit in the left image, the corresponding point can be found in the right image.

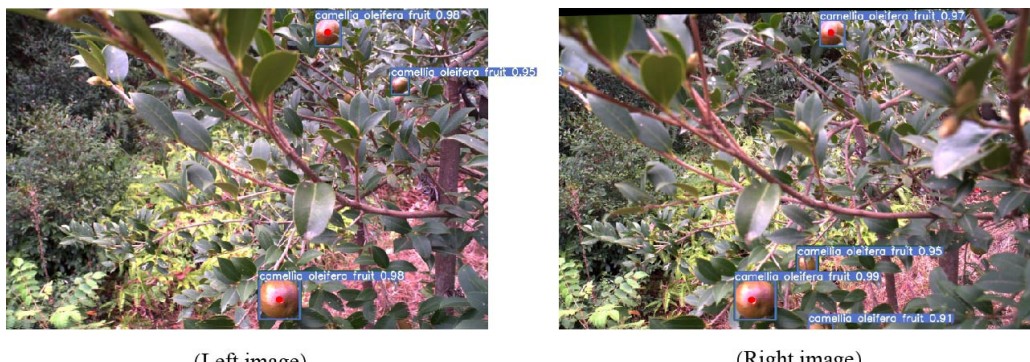

(Left image)  (Right image)

**Figure 10.** Matching effect of *C. oleifera* fruits under sunlight conditions. (**Left image**) is the left camera image, and (**right image**) is the right camera image in the binocular stereo vision system. The blue box is the area of LBP image texture matching, and the red dot is the center point of *C. oleifera* fruit obtained after matching.

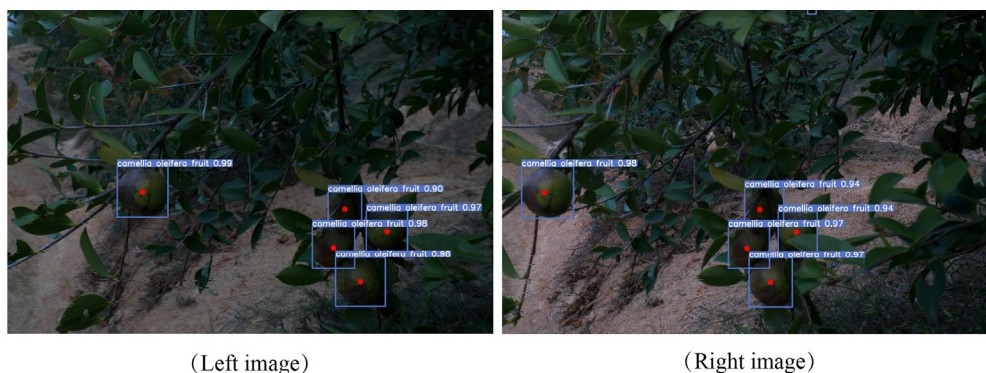

（Left image)  （Right image)

**Figure 11.** Matching effect of *C. oleifera* fruits under weak light condition. (**Left image**) is the left camera image, and (**right image**) is the right camera image in the binocular stereo vision system. The blue box is the area of LBP image texture matching, and the red dot is the center point of *C. oleifera* fruit obtained after matching.

Four *C. oleifera* fruits with different illumination and shading degrees were selected for testing, and the maximum diameter of the fruit was measured five times using a vernier caliper, with the average value used to compensate the measurement value of the laser rangefinder. The experimenter controlled the distance measuring device to measure 10 groups of data within the range of 400–600 mm from the fruit and calculated the error and standard deviation according to the experimental data as the evaluation standard. The experimental results are shown in Table 5 and Figure 12.

According to the results, the positioning accuracy of the binocular stereo vision system was slightly reduced in the orchard field test, but the error fluctuation at different points was small, which was close to the fluctuation trend of the experimental results on the indoor positioning accuracy of the binocular stereo vision system. In the experiment, the maximum error was 4.279 mm, the minimum error was 0.559 mm, the experimental standard deviation was 1.142 mm (the standard deviation of the distance measurement error statistics of all test fruits), and the average error of the experiment was 2.071 mm (the average error of the distance measurement error statistics of all test fruits).

**Table 5.** Statistics of the ranging errors in the orchard.

| Fruit 1 | | | Fruit 2 | | | Fruit 3 | | | Fruit 4 | | |
|---|---|---|---|---|---|---|---|---|---|---|---|
| Measurement Values/mm | Real Values/mm | Absolute Value of Error/mm | Measurement Values/mm | Real Values/mm | Absolute Value of Error/mm | Measurement Values/mm | Real Values/mm | Absolute Value of Error/mm | Measurement Values/mm | Real Values/mm | Absolute Value of Error/mm |
| 402.588 | 402 | 0.588 | 407.695 | 409 | 1.305 | 417.166 | 414 | 3.166 | 419.354 | 418 | 1.354 |
| 412.537 | 411 | 1.537 | 430.052 | 433 | 2.948 | 430.727 | 430 | 0.727 | 449.692 | 448 | 1.692 |
| 438.867 | 438 | 0.867 | 470.623 | 472 | 1.377 | 450.293 | 452 | 1.707 | 475.559 | 475 | 0.559 |
| 453.269 | 452 | 1.269 | 490.684 | 492 | 1.316 | 471.795 | 473 | 1.205 | 500.251 | 496 | 4.251 |
| 458.737 | 458 | 0.737 | 515.419 | 517 | 1.581 | 483.054 | 485 | 1.946 | 520.364 | 521 | 0.636 |
| 459.869 | 458 | 1.869 | 536.142 | 533 | 3.142 | 556.961 | 553 | 3.961 | 540.336 | 542 | 1.664 |
| 489.375 | 487 | 2.375 | 549.279 | 545 | 4.279 | 586.748 | 590 | 3.252 | 566.849 | 566 | 0.849 |
| 515.174 | 513 | 2.174 | 563.983 | 567 | 3.017 | 600.135 | 598 | 2.135 | 590.793 | 592 | 1.207 |
| 547.412 | 544 | 3.412 | 575.975 | 572 | 3.975 | 603.923 | 603 | 0.923 | 614.576 | 611 | 3.576 |
| 587.386 | 585 | 2.386 | 594.847 | 594 | 0.847 | 628.874 | 625 | 3.874 | 630.859 | 634 | 3.141 |
| Standard deviation | | 0.847 | Standard deviation | | 1.172 | Standard deviation | | 1.139 | Standard deviation | | 1.235 |
| correlation coefficient $R^2$ | | 0.9991 | correlation coefficient $R^2$ | | 0.9985 | correlation coefficient $R^2$ | | 0.9989 | correlation coefficient $R^2$ | | 0.9990 |

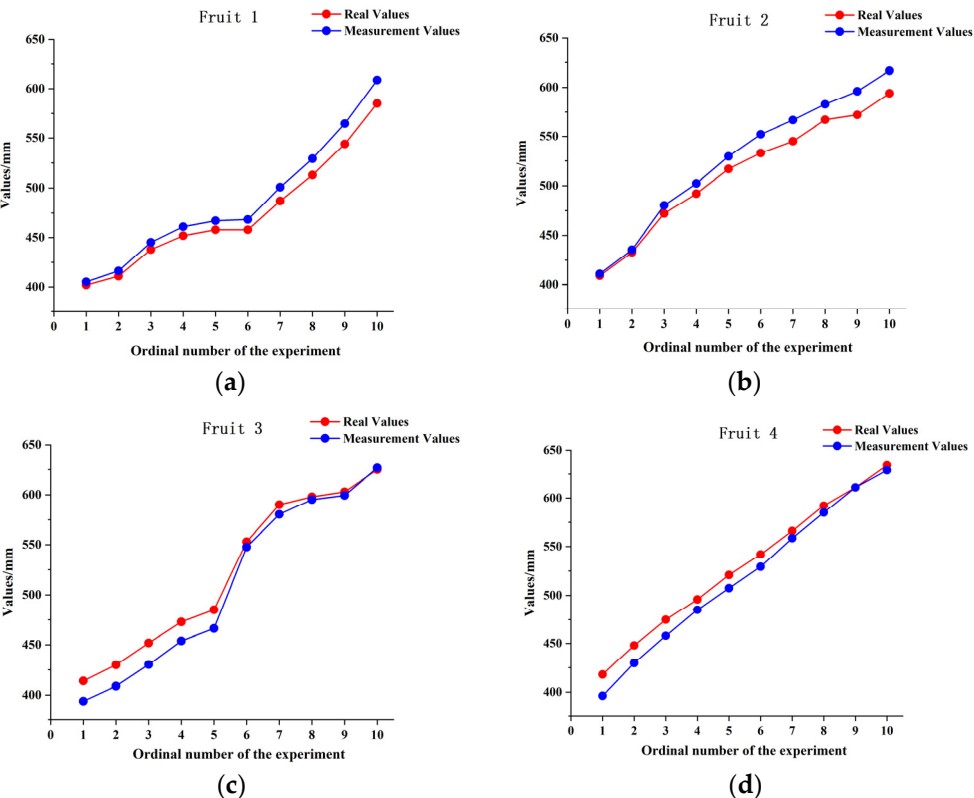

**Figure 12.** Regression plot of measurement value and real value: (**a**) regression plot of the experimental data of Fruit 1; (**b**) regression plot of the experimental data of Fruit 2; (**c**) regression plot of the experimental data of Fruit 3; (**d**) regression plot of the experimental data of Fruit 4.

## 4. Discussion

### 4.1. Analysis of the Recognition Results of C. oleifera Fruit

In this study, the best weight model after 300 rounds of iterative training using the YOLOv7 algorithm was selected as the weight model of the target detection algorithm for *C. oleifera* fruit recognition, and the confidence threshold of the target detection algorithm was set to 0.7. The experimental results show that, with regard to different occlusion conditions, slight occlusion had little effect on the recognition of *C. oleifera* fruits. However, under conditions in which some *C. oleifera* fruits were multi-shielded by leaves, branches or other fruits, the semantic information was seriously lost and difficult to recognize. For the *C. oleifera* fruits that were multi-shielded but identifiable, the detection model could accurately recognize the fruits in the unshielded range. Under sunlight conditions, part of the *C. oleifera* fruit surface was covered by the shadow of branches and leaves, which was obviously different from the uncovered area, resulting in recognition difficulty in the image. In future research, we will increase the *C. oleifera* fruit images under the conditions of occlusion and sunlight in the training set to improve the recognition accuracy of the detection model for *C. oleifera* fruit in natural environments.

### 4.2. Analysis of the Location Results of C. oleifera Fruit

To solve the problem that the anchor frame region of *C. oleifera* fruit obtained using the YOLOv7 target detection algorithm was deviated from the circumscribed rectangle of *C. oleifera* fruit outline and the difficulty in acquiring the actual center point of *C. oleifera* fruit by the anchor frame center only, this study obtained the coordinates of the same point on the surface of *C. oleifera* fruit in the left and right images through image template-matching of the LBP images of the anchor frame center region. Image-matching in the central region can improve the matching success rate of the center point and avoid mismatching due

to the loss of some pixels or the error of left and right image alignment. In the indoor localization experiment, the errors of the binocular stereo vision system in the *X*, *Y* and *Z* axes were close, and the maximum comprehensive error was 4.06 mm. It was feasible to evaluate the stability of the binocular stereo vision system by using the depth data of *C. oleifera* fruit positioning data in the natural field environment. Most of the data of the same fruit had little difference, and only a small part of the data had a great difference, which may be caused by the slight shaking of the manual handheld ranging device being difficult to avoid.

## 5. Conclusions

This study was dedicated to expanding the practical application of binocular stereo vision in the detection and positioning of *C. oleifera* fruits and to improving the robustness of the binocular stereo vision system in natural environments. The YOLOv7 algorithm was used to train the *C. oleifera* fruits data set, and the obtained detection model was used to recognize *C. oleifera* fruits. The test accuracy rate *P* of the model was 97.3%, the recall rate *R* was 97.6%, the *mAP* was 97.7% and the comprehensive evaluation index $F_1$ was 97.4%. The recognition experiments showed that the recognition rate of the model reached 93.13% under the condition in which the *C. oleifera* fruits were slightly shielded, the recognition rate reached 75.21% under the condition in which the *C. oleifera* fruits were seriously shielded. The recognition rate of the model was 90.64% under sunlight condition, and the recognition rate of the model was 91.34% under shading condition.

The binocular stereo vision system was built on the basis of an industrial camera to carry out three-dimensional space positioning on the *C. oleifera* fruits. The anchor frame regions of the *C. oleifera* fruits were matched by applying epipolar line constraint and sequence consistency principles, and template-matching was carried out on the LBP maps of the central regions of the left and right anchor frame images of the same *C. oleifera* fruits. Compared with the full-image matching or multi-feature point matching of the traditional stereo matching algorithm, the calculation and the mismatching rate of the localization algorithm amount in the stereo matching process can be effectively reduced. Compared with the commonly used gray matching in one-dimensional direction, the localization algorithm has higher positioning accuracy. The indoor positioning experiment showed that the maximum comprehensive error was 4.06 mm, the minimum comprehensive error was 0.37 mm and the average comprehensive error was 2.306 mm in the depth range of 400–550 mm.

In the orchard field experiment, the binocular stereo vision system had a maximum ranging error of 4.279 mm and an experimental standard deviation of 1.142 mm in the field of view with a depth of 400–600 mm, which basically met the working requirements of the precision picking robot. The binocular stereo vision system can be used for identifying and positioning fruits that are difficult to identify using traditional machine vision and can be applied to a harvesting robot for accurate picking, so the system has wide applicability. In future research, we will further study the activity range of the center point of *C. oleifera* fruits in the left and right images, reduce the size of the region to be matched and improve the running speed of the positioning algorithm. After completing this study, the binocular stereo vision system will be combined with the practical application of a *C. oleifera* fruit-picking robot.

**Author Contributions:** Conceptualization, X.L. and Y.X.; methodology, X.L. and Y.X.; software, X.L.; validation, X.L., Z.T. and S.C.; formal analysis, X.L.; investigation, X.L., Z.T. and A.L.; resources, Y.X. and Y.L.; data curation, X.L.; writing—original draft preparation, X.L.; writing—review and editing, Y.X.; visualization, X.L.; supervision, Y.X. and M.W.; project administration, Y.X.; funding acquisition, Y.X. All authors have read and agreed to the published version of the manuscript.

**Funding:** This research was funded by the National Key Research and Development Plan Project (2022YFD2002003).

**Data Availability Statement:** All data are presented in this article in the form of figures and tables.

**Conflicts of Interest:** The authors declare no conflict of interest.

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
