# Peer review of "Detection and Positioning of Camellia oleifera Fruit Based on LBP Image Texture Matching and Binocular Stereo Vision"

_agronomy, doi:10.3390/agronomy13082153_

Round 1
Reviewer 1 Report
● The papers titled “Detection and Positioning of Camellia Oleifera Fruit Based on LBP Image Texture Matching and Binocular Stereo Vision” states that the fruit tree is planted with plantation area reaching 4451 hectares of land in the in the hilly areas in the south of China.
● The use of traditional machinery carries the risk of buds being damaged affecting the yield in the following year due to flower and fruit synchronization characteristic features.
● There thus lies the need of precise picking for accurate identification and precise positioning of the Camellia Oleifera through using target detection algorithms employing YOLOv7
● Binocular stereo vision (Figure 1) and RGB-D (Red Green Blue and Depth) vision are at present the most commonly used three-dimensional spatial information acquisition technologies.
● Figure 2 shows Center point coordinate extraction process: (a) match the anchor frame regions; (b) cut the 155 anchor frame regions; (c) gray-scale image processing; (d) local binary patterns (LBP) maps; (e) set 156 the template region and the region to be matched; (f) result of template matching; (g) extract the 157 center point coordinates of binocular images.
● The Binocular stereo-vision ranging schematic diagram of Figure 2 for which YOLOv7 algorithm has been selected as the weight model of target detection algorithm for Camellia oleifera fruit recognition, and the confidence threshold of the target detection algorithm was set to 0.7.
● To further verify the performance of the binocular stereo vision system in a natural environment, a detection and positioning experiment has been conducted in a Camellia oleifera fruits planting base using a portal experimental device (KOMAX K-50) as shown in Figure 8 with schematic diagrams shown in Figure 9
● With regards to different occlusion conditions, the experimental results have shown that the slight occlusion had little effect on the recognition of Camellia Oleifera olleria, however the semantic information gets faded as the fruit is multi-shielded by leaves, branches or other fruit leaves.
● The test accuracy rate of the model has been shown to be 97.3%, the recall rate is 97.6%, the mAP is 97.7%, and the comprehensive evaluation index has been 97.4%. The recognition experiments have shown that the recognition rate of the model has reached 94.44% under the condition in which the Camellia oleifera fruits were slightly shielded, the recognition rate reached 70.24% under the condition in which the Camellia oleifera fruits were seriously shielded. The recognition rate of the model has been 89.65% under sunlight condition, and the recognition rate of the model was 93.44% under shading condition.
Is the subject matter presented in a comprehensive manner?
● The experiments have been carried out under strong and week light conditions as shown in Figure 10 and Figure 11, showing clearly that the corresponding point can be found as is evident in the right image of Figure 11.
● Four Camellia oleifera fruits with different illumination and shading degrees were selected for testing, and the maximum diameter of the fruit was measured five times using a vernier caliper, with the average value used to compensate the measurement value of the laser rangefinder.
● The experimenter controlled the distance measuring device to measure 10 groups of data within the range of 400–600 mm from the fruit, and calculated the error and standard deviation according to the experimental data as the evaluation standard. The experimental results are shown in Table 5 for four types of fruit categories.
● The orchard field experiment, the binocular stereo vision system had a maximum ranging error of 4.279 mm and an experimental standard deviation of 1.142 mm in the field of view with a depth of 400–600 mm, which basically met the working requirements of the precision picking robot.
Are the references provided applicable and sufficient?
· The authors take support from thirty two (32) current and branded references including science direct and agronomy, comparing ultimately result with [41]
Author Response
Dear Editors and Reviewer:
Thank you very much for your comments on our manuscript entitled ‘Detection and positioning of Camellia oleifera fruit based on LBP image texture matching and binocular stereo vision’ (ID: agronomy-2472758). These comments are very valuable for the revision and improvement of our paper, and also have important guiding significance for our research. We have carefully studied comments and made corrections, and hope to get your approval. In addition, the changed parts in the revised manuscript can be easily found highlighted in red.
We would like to thank the anonymous reviewers and the editor for their constructive criticism that allowed us to improve our paper, and we believe that the quality of this work has significantly improved.
Thanks again to the reviewers and editors for your comments and suggestion. If you have any queries, please don’t hesitate to contact me at the address below.
Thank you and best regards.
Yours sincerely,
Xiangming Lei
E-mail: lxm@stu.hunau.edu.cn

Reviewer 2 Report
The article presents an acceptable scientific level and the scope of the research and its results bring new knowledge to agricultural engineering, especially in the area of vision identification and automated fruit picking in a selective manner.
Author Response

(The authors gave the same response as above.)

Reviewer 3 Report
The articule is well written. I have some comments:
1. Due to the YOLOv7 Algorithm, was used. It's necessary a section with a brief explain of the Algorithm.
2. For clarity of the presentation, I suggest to include also a image where the detection did not work. (Figures 4 and 10)
3.- You must to include the results of the training, validation as test dataset.
4.- The table 5 it's difficult to read. I suggest to include also a regression plot (real value vs measurement value) and include the value of correlation coefficient
Author Response
Dear Editors and Reviewer:
Thank you very much for your comments on our manuscript entitled ‘Detection and positioning of Camellia oleifera fruit based on LBP image texture matching and binocular stereo vision’ (ID: agronomy-2472758). These comments are very valuable for the revision and improvement of our paper, and also have important guiding significance for our research. We have carefully studied comments and made corrections, and hope to get your approval. In addition, the changed parts in the revised manuscript can be easily found highlighted in red.
The main corrections in the manuscript and the responds to the comments are as follows:
- Due to the YOLOv7 Algorithm, was used. It's necessary a section with a brief explain of the Algorithm.
Response 1: Thank you very much for your suggestions on our manuscript. In the manuscript of revised version, We have added a brief explanation of Yolov7 algorithm on Line 116 to 124, Page 3.
- For clarity of the presentation, I suggest to include also a image where the detection did not work. (Figures 4 and 10)
Response 2: Thank you very much for your suggestions on our manuscript. I'm sorry that we didn't find any image that didn't work at all. But we found some images which include part of the detection did not work. So we have replaced the image (Figures 4 and 10) which include part of the detection did not work on Line 230, Page 7 and Line 337, Page 11.
- You must to include the results of the training, validation as test dataset.
Response 3: Thank you very much for your suggestions on our manuscript. I'm sorry that we didn't think about that before. We have modified source of the test dataset, the test dataset was extracted from the training set and the verification set, and the statistical experiment was re-conducted. Please refer to the revised manuscript for the new experimental results on Page 7.
- The table 5 it's difficult to read. I suggest to include also a regression plot (real value vs measurement value) and include the value of correlation coefficient
Response 4: Thank you very much for your suggestions on our manuscript. We have added the regression plot of the experimental data (Figures 12) on Line 351 to 354, Page 12. The values of correlation coefficient have been added to Table 5 on Page 12.
The above are my main corrections to the paper and the responses to the Reviewer’s comments. We would like to thank the anonymous reviewers and the editor for their constructive criticism that allowed us to improve our paper, and we believe that the quality of this work has significantly improved.
Thanks again to the reviewers and editors for your comments and suggestion. If you have any queries, please don’t hesitate to contact me at the address below.
Thank you and best regards.
Yours sincerely,
Xiangming Lei
E-mail: lxm@stu.hunau.edu.cn
